# Spontaneous Resolution of an Aggressive Direct Carotid Cavernous Fistula Following Partial Transvenous Embolization Treatment: A Case Report and Review of Literatures

**DOI:** 10.3390/medicina60122011

**Published:** 2024-12-05

**Authors:** Wen-Jui Liao, Chun-Yuan Hsiao, Chin-Hsiu Chen, Yuan-Yun Tseng, Tao-Chieh Yang

**Affiliations:** 1Department of Neurosurgery, Chung Shan Medical University Hospital, Taichung City 402, Taiwan, China; cshy660@csh.org.tw (W.-J.L.); cshy2140@csh.org.tw (C.-H.C.); 2Department of Medical Education, Chung Shan Medical University Hospital, Taichung City 402, Taiwan, China; cyhs1504@gmail.com; 3Department of Neurosurgery, New Taipei Municipal Tu-Cheng Hospital (Built and Operated by Chang Gung Medical Foundation), New Taipei City 236, Taiwan, China; britsey@tmu.edu.tw; 4School of Medicine, Chung Shan Medical University, Taichung City 402, Taiwan, China

**Keywords:** carotid cavernous fistulas, internal carotid artery, cortical vein reflux, endovascular treatment, digital subtraction angiography, magnetic resonance imaging

## Abstract

Traumatic direct type carotid cavernous fistula (CCF) is an acquired arteriovenous shunt between the carotid artery and the cavernous sinus post severe craniofacial trauma or iatrogenic injury. We reported a 46-year-old woman who had developed a traumatic direct type CCF after severe head trauma with a skull base fracture and brain contusion hemorrhage. The clinical manifestations of the patient included pulsatile exophthalmos, proptosis, bruits, chemosis, and a decline in consciousness. Magnetic resonance imaging (MRI) revealed engorgement of the right superior ophthalmic vein (SOV), perifocal cerebral edema in the right frontal–temporal cortex, right basal ganglia, and brain stem. Digital subtraction angiography (DSA) disclosed a direct type high-flow CCF with an aggressive cortical venous reflux drainage pattern, which was attributed to Barrow type A and Thomas classification type 5. After partial treatment by transvenous coil embolization for the CCF, the residual high-flow fistula with aggressive venous drainage had an unusual rapid spontaneous resolution in a brief period. Therefore, it is strongly recommended to meticulously monitor the clinical conditions of patients and perform brain MRI and DSA at short intervals to determine the treatment strategy for residual CCF after partial endovascular treatment.

## 1. Introduction

A carotid cavernous fistula (CCF) is an abnormal connection between the cavernous sinus and the internal and external carotid arteries (ICA and ECA), or any of their branches. Several classification schemes have categorized CCFs according to etiology (traumatic or spontaneous), angiographic arterial architecture (direct or indirect), venous drainage pattern (anterior, posterior, inferior with/without retrograde drainage into cortical vein), or hemodynamic features (high flow versus low flow) [1]. Barrow et al. distinguished four types of CCFs (type A to D) based on arterial angioarchitecture [2], and Thomas et al. proposed a five-type classification system (type 1 to 5) based on the angioarchitecture of the venous phase [3]. Post-traumatic CCFs, predominantly of Barrow type A, comprise approximately 75–80% of high-flow direct fistulas between the ICA and the cavernous sinus [1]. Patients with CCFs often exhibit Dandy’s triad, which includes pulsatile exophthalmos, chemosis, and bruit [4,5]. Other uncommon neurological deficits include cranial nerve palsy and deterioration of consciousness. Most daunting is cortical symptomatology such as brain hemorrhage, perifocal edema, and seizures resulting from retrograde drainage into the superficial middle cerebral vein (SMCV) or the petrosal vein at the posterior fossa [3,6,7]. Spontaneous closure of CCFs can be found in indirect or low-flow CCFs, but is uncommon in post-traumatic high-flow CCFs [8,9]; thus, prompt treatment for post-traumatic CCFs is often necessary to restore ocular function and normal cerebral circulation [5,10]. In the literature, we report a traumatic direct CCF case with an aggressive venous drainage pattern, which was classified as Barrow type A [2] and Thomas classification type 5 [3]. After partial treatment with transvenous coil embolization, the residual high-flow fistula with aggressive cortical venous drainage persisted; however, the angiographic feature and perifocal edema of the basal ganglia and brain stem were rapidly resolved. Therefore, no further endovascular or surgical treatments were carried out for the patient.

## 2. Case Report

A 46-year-old woman with medically controlled hyperthyroidism was a victim of a motor vehicle accident on the road. The accident led to poor consciousness with Glasgow Coma Scale (GCS) level at E4V1M4, nasal bleeding, and left otorrhea in the patient. A brain computed tomography (CT) scan (Figure 1) showed a small subdural hematoma over the right temporal lobe, right Sylvian subarachnoid hemorrhage, and a right sellar floor fracture with sphenoid sinus hematoma upon arrival at the emergency room. The patient underwent right intracranial pressure monitor insertion for further monitoring and intensive care. Fortunately, the severity of traumatic brain injury did not progress. She gradually regained consciousness 3 days post-trauma. However, she complained of a progressive right eye floater and flash 4 weeks later, so she visited an ophthalmologist for further evaluation. Initial examination revealed right eye proptosis and conjunctival vessels engorgement, without chemosis, or tinnitus in the right ear. (Figure 2a) Rapidly progressive severe right eye exophthalmos, audible bruits, and chemosis (Figure 2b) developed within several days, highly suggesting right carotid cavernous fistula. The visual acuity and intraocular pressure data before treatment and one month post-treatment of the CCF are listed in Table 1. In addition, she became unconscious (GCS E3V2M5) and had left limb hemiplegia. A brain magnetic resonance imaging (MRI) (Figure 3a) disclosed engorgement of the right SOV and swelling of right orbital cavity (the MRI protocol we used is listed below in Table 2). In addition, high T2-weighted imaging signal changes were observed in the right corpus striatum (Figure 3b), right insular lobe, right inferior frontal lobe, right medial temporal lobe, midbrain, and pons (Figure 3c), which indicate perifocal brain edema.

Digital subtraction angiography (DSA) was performed to confirm a direct type high-flow CCF of the right-side ICA with reflux flow into the right SOV and inferior ophthalmic vein (IOV), the right superficial middle cerebral vein (SMCV), the right superior petrosal sinus (SPS), and the right inferior petrosal sinus (IPS). Right facial vein engorgement due to SOV/IOV drainage was also noticed in the DSA (Figure 4a,b). Therefore, the CCF was classified as Barrow type A based on the arterial angioarchitecture [2] and Thomas classification type 5 based on the venous drainage pattern classification [3].

The neurointerventionists performed transvenous endovascular embolization 3 days after DSA via the right facial vein to the SOV route for the CCF, due to the initial failure of the right IPS route because of tortuous and stenotic right IPS.

A total of 28 Guglielmi Detachable Coils (GDCs) were deployed into the CCF, resulting in occlusion of the anterior drainage part (SOV/IOV) of CCF. However, there were still two fistulous points in the posterior part of the CCF showing severe arteriovenous shunts. The first was located at the right sphenoparietal sinus, draining into the right SMCV. The second was at the right SPS, which caused severe cortical vein reflux (CVR) (Figure 4c,d).

Brain MRI was followed up 3 days after endovascular treatment, which disclosed significantly decreased engorgement of the right SOV (Figure 5a) and less perifocal edema in the right corpus striatum, midbrain, and pons (Figure 5b,c). Consequently, a conservative strategy was implemented to address residual CCF. We conducted a follow-up DSA one month after the TVE. The DSA revealed that the residual CCF had spontaneously resolved without additional cortical reflux (Figure 6a,b). The patient experienced a complete remission of ophthalmic symptoms (Figure 2c); her conscious level and the intraocular pressure of her right eye (Table 1) both had a favorable recovery.

## 3. Discussion

CCFs can be classified as direct or indirect type simply based on the arteriovenous shunt angioarchitectures. Direct fistulas have an abnormal communication between the ICA and the cavernous sinus. Indirect fistulas have an abnormal communication between the meningeal branches of the ICA and ECA and the cavernous sinus [11]. Based on the flow rate, CCFs are categorized into high (where all blood from the ICA enters the fistula without perfusing intracranial vessels), intermediate (where both the fistula and intracranial vessels receive blood from the ICA), and low flow (where only a slow and sluggish filling of the cavernous sinus is observed) [2,8]. Direct CCFs are often attributable to trauma, fibromuscular dysplasia, intracavernous artery aneurysms rupture, collagen insufficiency, arterial dissection, or iatrogenic factors (e.g., surgical trauma) [4,5]. CCFs permit the direct transmission of highly pressurized arterial blood into the cavernous sinus and its draining veins, resulting in venous hypertension. The clinical manifestation of CCF results directly from increased intracavernous pressure and altered flow dynamics. The altered venous drainage of CCFs may be directed towards the ophthalmic venous system anteriorly; the SPS, the IPS, or the basilar plexus posteriorly; the sphenoparietal sinus laterally; the intercavernous sinus contralaterally; and the pterygoid plexus via the vein of the foramen rotundum and the vein of the foramen ovale inferiorly. The venous drainage is often multidirectional [1,3]. Cortical venous reflux (CVR) of CCFs is defined as arterial blood filling of the superficial middle cerebral veins, perimesencephalic, and cerebellar venous systems. The high-pressure blood flow of CVR reverses into the sphenoparietal sinus, or the posterior fossa via the petrosal vein with occlusion of other drainage pathways, resulting cerebral cortical venous hypertension [3,12]. The impairment of normal cerebral venous drainage by CVR induces intracranial venous hypertension and leads to ICH, brain perifocal edema, or cerebral venous infarction [6,10].

The majority of CCF patients exhibit proptosis (90%), chemosis (90%), diplopia (50%), pain (25%), trigeminal nerve dysfunction, visual impairment (up to 50%), and ICH (5%) [1,5]. Treatment for traumatic direct CCFs, especially those with aggressive cortical venous drainage, is often prompt because of worsening ocular symptoms (e.g., exophthalmos or visual acuity loss) and neurologic deficits (e.g., hemiplegia or conscious decline related to brain edema or hemorrhage) [4,5].

The primary treatment for traumatic high-flow direct CCF in contemporary practice is endovascular therapy, which occludes the fistula while preserving the parent ICA by transarterial or transvenous approaches or combination together [1,13,14]. In the 1980s, transarterial balloon embolization of direct CCFs became the standard method [15]. However, due to the limited commercial availability of detachable balloons, including in Taiwan, detachable coils have become a widely employed endovascular tool for the treatment of direct CCFs in many countries. The advantages of coil applications are their easy retrievability, ability to be repositioned, and better control [14]. Following recent advances in endovascular treatment, liquid embolic materials and device technology, such as covered stents, or flow-diverting stents have been applied for the treatment of direct CCFs [14,16]. The utility of liquid embolic materials, such as histoacryl glue (N-butyl-2 cyanoacrylate n-BCA) and Onyx (EVOH: ethylene vinyl alcohol), for direct CCFs was recently reported in several studies [17,18,19]. Sufficient skill and experience are required to achieve complete occlusion of high-flow direct fistulas without unintentional glue reflux or distal migration, which may lead to cranial nerve palsies, trigeminal cardiac reflux, or ICA occlusion [14,20]. Recent advances in endovascular techniques such as placement of polyfluorotetra ethylene-covered stents have created alternatives to ICA sacrifice in traumatic arterial damage, especially in the setting of an unsuccessful balloon occlusion study. Covered stent grafts have the technical disadvantage of limited longitudinal flexibility, making it difficult to navigate them through the tortuosity of the cerebral arteries. The complications of covered stents or flow diverting stents include endoleak, coverage of vital perforators, dissection, and rupture [14,20,21]. The targeted treatment of CCFs with single or overlapping flow-diverting devices (FDDs), with or without supplementary embolic agents, has demonstrated a high success rate for both clinically and in long-term angiographic outcomes compared to other standalone endovascular techniques. Nevertheless, multi-center prospective studies require more investigation [22,23]. In our case, we adopted GDC (Boston Scientific Corporation, Boston, MA, USA) embolization via a transvenous route due to the lack of a detachable balloon in Taiwan and the higher obliteration rate in the complex venous drainage pattern of the CCF [14,16].

Prompt, complete treatment to obliterate the outlets of the cavernous sinus to dangerous venous drainage systems (CVR, deep venous drainage, anterior drainage) is the major goal for aggressive direct CCFs to avoid visual loss, cranial nerve palsy, intracranial hemorrhage (ICH), or brain perifocal edema [1,10,24].

The prognosis of CCFs after endovascular treatment is generally good but may have eccentric features, including paradoxical deterioration, recurrence after embolization, and enduring symptoms [25,26]. Unintended partial embolization of the posterior portion of a CCF can cause high-flow drainage retrograde in the SMCV, posterior fossa, or pontomedullary venous system, which might intensify symptoms or create life-threatening complications, such as brain stem congestion or intracranial hemorrhage (ICH) due to increased venous pressure [6,7,26,27].

There are still controversies about how to manage partial treatment of CCFs, especially residual aggressive type CVR [26,28,29]. Monitoring angiographic changes after partial treatment of CCFs is crucial. Spontaneous thrombosis of residual fistulas of CCFs post incomplete treatment is common and especially seen in low-flow, small-sized fistulas, hypotension, severe ocular manifestations, dissections or spasms of the carotid artery, and increased intracranial pressure due to ICH [8,9,12,29,30]. Residual shunts and the late-restrictive types were associated with poor recovery [25]. If residual CCFs still resulted in high-flow CVR with corpus striatum or brain stem edema, the management strategy needed to be very cautious [27,31]. The spontaneous complete regression of aggressive CCFs is an extremely uncommon phenomenon [8,9]. Some reports such as Lee et al. and Mochizuki et al. showed catastrophic ICH post partial endovascular treatment of CCFs and therefore advocated against a “wait and see” strategy in the residual aggressive CCFs [27,31]. After the initial partial endovascular treatment, some patients require additional interventions for residual CCFs, such as a second endovascular treatment or Gamma Knife Radiation [5,24,26].

The mechanism of spontaneous closure of the high-flow residual CCF remains unclear. It has been proposed that the process of cure of CCFs may result in thrombosis of the cavernous sinus [8,9]. After reviewing the literature, several possible mechanisms have been proposed for the spontaneous closure of CCFs. There are several factors that are related to spontaneous resolution of CCFs: (1) small size of fistula; (2) hypotension such as anaphylactic shock or hemorrhage; (3) venous stasis and damaged orbital tissue due to decreased arteriovenous fistula gradient between the ICA and cavernous sinus; (4) dissection or spasm of carotid artery during DSA or endovascular treatment; (5) increased intracranial pressure due to ICH or infarction; (6) contrast media during DSA or endovascular treatment; and (7) absence of posterior drainage. [8,32,33] Although several mechanisms may be involved in the spontaneous closure of a CCF, neither its occurrence nor the time frame is predictable.

In our case, initial, high-flow CVR post partial treatment of the CCF was found after occlusion of the anterior drainage part via the trans-SOV route. We followed up with brain MRI soon after endovascular treatment. Given the improvement in clinical symptoms and the reduced perifocal edema of the corpus striatum and brain stem on the follow-up brain MRI, we decided to closely monitor the change in the residual CCF and refrain from further endovascular or surgical treatment. After implementing a conservative strategy for the residual CCF for one month, the follow-up DSA showed spontaneous thrombosis of the residual CCF without any additional CVR. Analyzing the clinical course and angiographic features of our case, the fistula between the ICA and cavernous sinus founded by DSA was small in size. Spontaneous closure of CCFs may be more likely in cases of small-sized fistulas [8,9]. The patient had twice iodinated contrast material into cerebral vessels, especially feeding arteries and draining the veins of CCFs, one for diagnostic DSA, and one for the endovascular treatment via transvenous approaches. Thrombogenic properties of the contrast material of angiography probably induced spontaneous closure of the CCF. The possible mechanisms included a direct effect on the vascular endothelium, exaggerated leukocytic accumulation, and promoted RBC aggregation [8,33]. High-osmolar contrast media can cause direct damage to the endothelial cells lining the blood vessels. This injury can lead to an inflammatory response, which includes the upregulation of adhesion molecules on the endothelial surface. High-osmolar contrast agents also increase the osmolarity of the blood, which can lead to changes in blood viscosity and flow properties. This can promote leukocyte adhesion and aggregation. Contrast media can activate leukocytes, leading to an increase in the expression of adhesion molecules such as CD11b on their surface. This enhances their ability to adhere to the endothelium. The combination of endothelial injury, increased leukocyte adhesion, and changes in blood flow can create a pro-thrombotic environment, leading to the formation of blood clots [33,34].

It is also proposed that the navigation of microguidewires and catheters through small-sized fistulas during DSA and attempted intervention could induce abrupt closure of fistulas (via arterial spasm or dissection), especially those supplied by small feeding arteries [8,9]. After partial endovascular treatment, the greatly reduced arteriovenous flow and the pressure gradient between the ICA and CCFs may lead to thrombosis of the feeding arteries, draining veins, and result in closure of the fistula as well [8,9,12].

According to the experience of our case, the residual aggressive high-flow CCFs after partial endovascular treatment had a chance of spontaneous resolution. Clinical ophthalmic conditions including intraocular pressure, visual acuity, and extraocular muscle function associated with the oculomotor, trochlear, and abducent nerves should be intensively monitored. We also need to pay attention to the state of consciousness, epilepsy, and increased intracranial pressure. Regarding the management strategy of partially treated CCFs, we advocate that the patient’s clinical conditions should be monitored first, rather than re-treating immediately. It is important to perform further MRI scans and DSA soon after to detect possible changes in cerebral parenchyma and blood flow changes in CCFs. If clinical neurological and ophthalmological symptoms worsen, prompt treatment of residual aggressive CCFs is necessary. Future investigations should focus on identifying the factors that contribute to the spontaneous resolution of residual CCFs after applying new endovascular technologies and materials such as Onyx and flow-diverting stenting [22,35].

## 4. Conclusions

Treating direct high-flow CCFs sometimes leaves residual lesions with CVR unintentionally. Based on our patient’s experience of spontaneous resolution of an aggressive direct CCF following partial transvenous treatment, it is crucial to closely monitor the clinical and ophthalmic conditions of patients first, then use a follow-up brain MRI and cerebral angiography closely to determine further treatment strategies of residual aggressive CCFs.

## Figures and Tables

**Figure 1 medicina-60-02011-f001:**
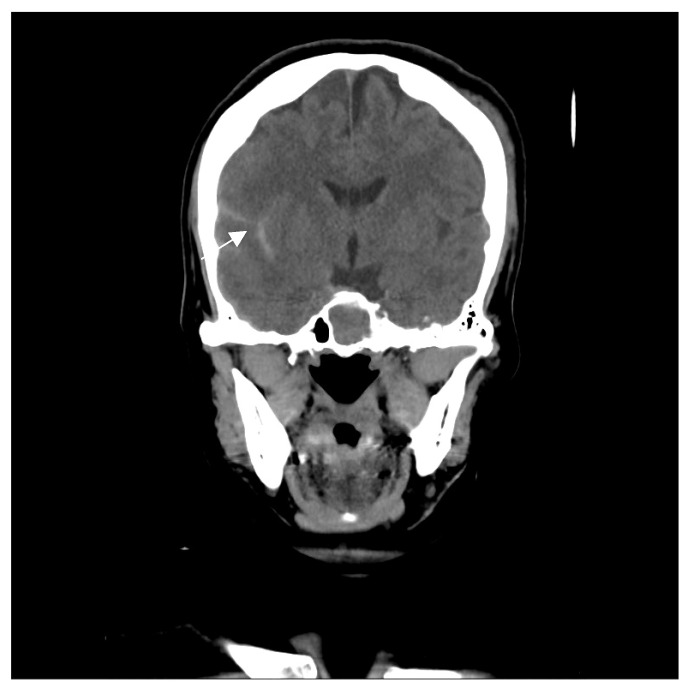
The figure reveals a small subdural hematoma over the right temporal lobe (arrow), sylvian subarachnoid hemorrhage, and right sellar floor fracture with sphenoid sinus hematoma.

**Figure 2 medicina-60-02011-f002:**
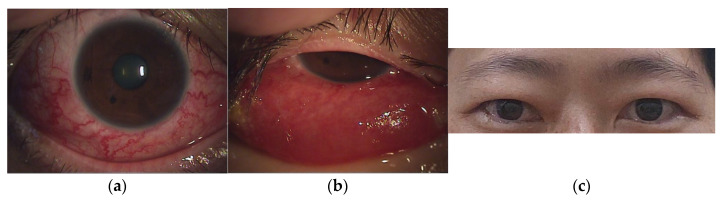
(**a**) Right eye proptosis and conjunctiva vessel engorgement, without chemosis. (**b**) Rapidly progressive severe right eye exophthalmos and chemosis developed in several days. (**c**) After 1 month of treatment, the right eye had a full recovery without strabismus.

**Figure 3 medicina-60-02011-f003:**
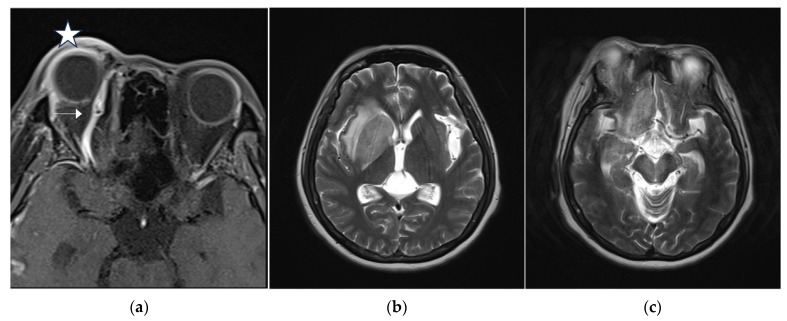
(**a**) Brain magnetic resonance imaging (MRI) axial view, T1-weighted imaging with contrast disclosed engorgement of right superior ophthalmic vein (arrow head), and swelling of right orbital cavity (star); (**b**) brain MRI axial view, T2-weighted imaging show high T2 signal change in right corpus striatum; and (**c**) axial view of brain MRI T2-weighted imaging show enhancement at right insular lobe, right frontal inferior lobe, right medial temporal lobe, midbrain, and pons, which indicate perifocal cerebral edema.

**Figure 4 medicina-60-02011-f004:**
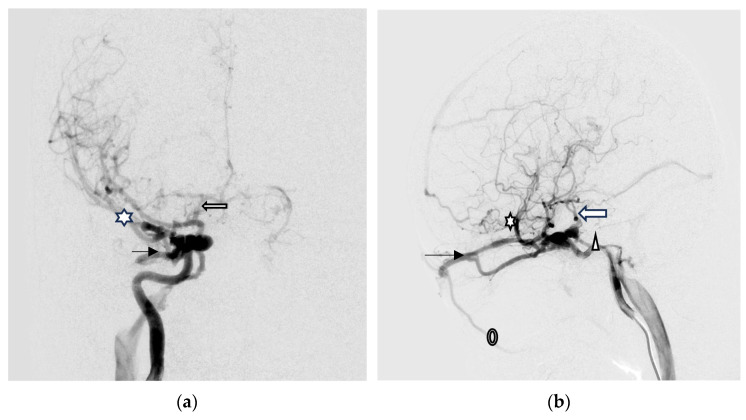
(**a**,**b**) Digital subtraction angiography (DSA) anterior–posterior view (AP) and lateral (Lat) view show a direct type high-flow CCF on the right-side ICA with reflux flow into the right superior/inferior ophthalmic vein (arrow head), the right superficial middle cerebral vein (star), the right superior petrosal sinus (arrow), and the right inferior petrosal sinus (triangular). Right facial vein engorgement due to right superior/inferior ophthalmic vein drainage was also noticed in the DSA (circle); (**c**,**d**) The Guglielmi Detachable Coils (GDCs) were deployed into the CCF, resulting in occlusion of the anterior drainage part of CCF (arrow head). There were still two fistulous points in the back part of the CCF that showed severe arteriovenous shunts: one at the right sphenoparietal sinus drainage into the right superficial middle cerebral vein (star), and another at the right superior petrosal sinus (arrow). This caused severe cortical vein reflux.

**Figure 5 medicina-60-02011-f005:**
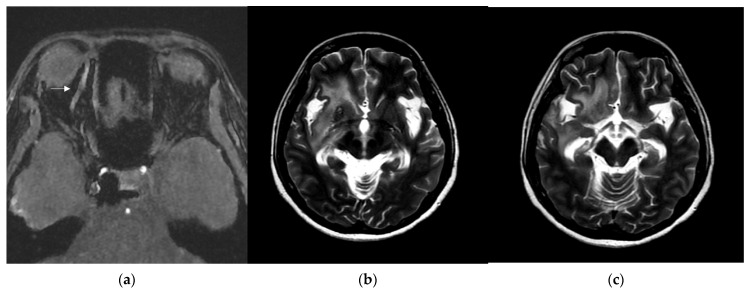
(**a**) Brain MRI was followed up 3 days after endovascular treatment: axial view of brain MRI T1-weighted imaging with contrast disclosed significantly decreased engorgement of the right superior ophthalmic vein (arrow head); (**b**,**c**) brain MRI axial view, T2-weighted imaging showed less perifocal edema in the right corpus striatum, midbrain, and pons comparing to the condition before endovascular treatment.

**Figure 6 medicina-60-02011-f006:**
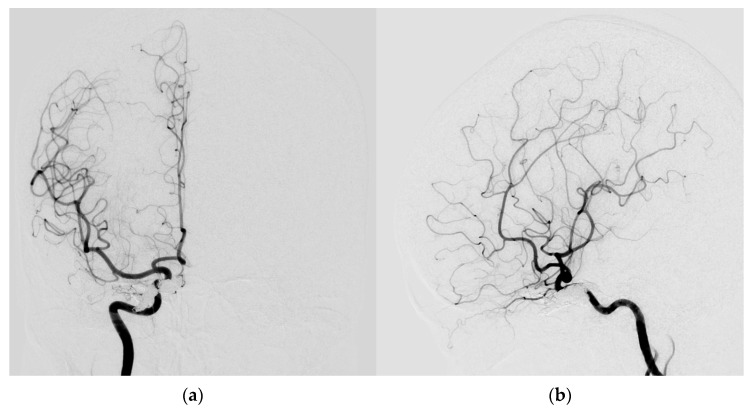
(**a**,**b**) The AP and Lat DSA views revealed the residual CCF had spontaneously resolved without additional cortical reflux.

**Table 1 medicina-60-02011-t001:** Visual acuity and intraocular pressure of the patient.

	OD (Right Eye)	OS (Left Eye)
Pre-treatment VA	0.04	0.8
Post-treatment VA	0.9	0.9
Pre-treatment IOP	31 mmHg	13 mmHg
Post treatment IOP	12 mmHg	11 mmHg

Visual acuity: VA; intraocular pressure: IOP.

**Table 2 medicina-60-02011-t002:** Brain magnetic resonance imaging with pulse sequence protocol.

Coronal cut: T2-weighted imaging Fast Recovery Fast Spin Echo (T2 frFSE) Axial cut: T2 frFSE, T1-weighted imaging Fluid-Attenuated Inversion Recovery (T1 FLAIR), T2-weighted imaging FLAIR Fat Saturation (T2 FLAIR fs) Diffusion-Weighted Imaging (DWI) b1000 3D TOF MRA: Three-dimensional Time of Flight Spoiled Gradient Recalled Echo Fat Saturation HyperSense ( 3D TOF SPGR FS HyperSense) Post-contrast: Contrast-Enhanced Fat Saturation Three-dimensional Sagittal T1-weighted imaging Cube (C+ fs 3D SAG T1 Cube), C+ AXI T1 SPGR, C+ SAG T1 SPGR CEMRA: Contrast-Enhanced Magnetic Resonance Angiography CEMRV: Contrast-Enhanced Magnetic Resonance Venography

## Data Availability

The clinical data (patient data, pictures, etc.) were all retrieved with the consent of the patients and the hospital involved. All referential data were retrieved with consent.

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
