# Peer review of "Spontaneous Resolution of an Aggressive Direct Carotid Cavernous Fistula Following Partial Transvenous Embolization Treatment: A Case Report and Review of Literatures"

_medicina, 2024, doi:10.3390/medicina60122011_

Round 1

Reviewer 1 Report

Comments and Suggestions for Authors

Abstract: The sentence, (Traumatic direct type carotid-cavernous fistulas (CCFs) are acquired arteriovenous shunts between the carotid artery and the cavernous sinus post head injury, particularly high energy head trauma, iatrogenic injury, or severe blunt craniofacial fracture) , is confusing due to its length and structure. I recommend rephrasing it for clarity.

Additionally, the phrase, (We reported a 46-year-old woman had developed a traumatic…) seems to have a missing word. Please revise. 

Overall, I suggest rewriting the first paragraph to improve readability and language flow.

Introduction: The section is clear, but it would benefit from an expanded description of the disease to provide readers with a fuller understanding of CCFs.

Case Report: In the phrase, (Figure 1) showed a small subdural hematoma, I recommend marking the hematoma with an arrow in Figure 1. This will help readers who may not be familiar with diagnostic imaging to identify it more easily.

In Figure 3, please add a description of the view and type of image (e.g., axial, sagittal, or coronal, and T1 or T2 weighted). This image is in an axial view, T2-weighted .

Discussion: The first sentence is repetitive of the first sentence in the abstract. Consider rephrasing for variety.

Minor Comments : 

Avoid repeating a term with its abbreviation each time; use the full form once and then the abbreviation consistently afterward.

The MRI protocol you used should be organized in a table format, as it is a significant component of your case report.

There are minor language errors; please review and correct them for overall clarity.

Author Response

Our reply had been listed in the attached file, please see the attachment.

Thank you !

Reviewer 2 Report

Comments and Suggestions for Authors

The article of Wen-Jui Liao et al. presents a case report that discusses an unusual phenomenon such as the spontaneous closure of a direct carotid-cavernous fistula (CCF) after partial embolization, particularly in a case involving vigorous venous draining. The report details the patient's presentation, diagnosis, imaging, and follow-up, which allow to comprehend the clinical reasoning and sequence of interventions. Here are some recommendations that could be beneficial for the article:

1. A more in-depth examination of the possible mechanisms underlying the spontaneous resolution reported in this example will improve this section. Although spontaneous resolution is the focus, the report might benefit from a more in-depth explanation of the potential mechanisms underlying this phenomena. Currently, the mechanisms are briefly discussed but not in sufficient depth.

2. Quantitative data on the patient's clinical and radiological progression would enhance the report's significance. For example, specific changes in proptosis, ocular motility, or vascular flow measurements following intervention would provide a more accurate picture of recovery.

3. Several abbreviations are used in the article (e.g., EVT for endovascular treatment, CCF for carotid-cavernous fistula), although they are not always clearly defined or used consistently. Please revise.

Author Response

(The authors gave the same response as above.)

Reviewer 3 Report

Comments and Suggestions for Authors

The manuscript provides a thorough discussion of traumatic direct carotid-cavernous fistulas (CCFs) and their management. However, there are areas where clarity and depth can be improved. First, the distinction between direct and indirect CCFs should be made earlier in the discussion, as this would help readers understand the pathophysiology of the condition more clearly. The term cortical venous reflux (CVR) is central to the discussion, but it would benefit from a more precise definition, particularly in explaining its role in venous congestion and neurologic complications such as brain edema and hemorrhage. A more detailed explanation of the advantages of coil embolization, especially in relation to liquid embolics and flow-diverting stents, would strengthen the manuscript’s treatment discussion. Furthermore, the section on spontaneous thrombosis of residual CCFs should be elaborated to clarify the mechanisms behind this phenomenon. Specifically, the role of angiographic contrast agents in promoting thrombosis, via endothelial injury and activation of the coagulation cascade, should be explained in more detail. Additionally, the distinction between low-flow and high-flow CCFs is critical when discussing the likelihood of spontaneous thrombosis and closure, as these two types may have different pathophysiological mechanisms. In terms of literature references, the mention of studies by Al-Afif, Shadi et al., and Lee et al. would benefit from a more thorough summary of their findings to contextualize how they contribute to the current treatment strategies. Lastly, the manuscript’s section on neurological and ophthalmic monitoring post-EVT should specify which cranial nerve functions, visual acuity, and intraocular pressure need to be closely monitored, to provide more actionable guidance for clinicians. Strengthening the conclusion by emphasizing personalized treatment strategies and future research directions would also enhance the manuscript’s impact, guiding further exploration into predictive factors for spontaneous resolution and more effective treatment protocols.

Author Response

(The authors gave the same response as above.)

Round 2

Reviewer 1 Report

Comments and Suggestions for Authors

The authors have addressed most of the questions . No further comments 

Reviewer 2 Report

Comments and Suggestions for Authors

I have not further comments. Articles could be accepted for publication